# Stereotactic Radiosurgery in Metastatic Spine Disease—A Systemic Review of the Literature

**DOI:** 10.3390/cancers16162787

**Published:** 2024-08-07

**Authors:** Adriana Palacio Giraldo, David Sohm, Johannes Neugebauer, Gianpaolo Leone, Marko Bergovec, Dietmar Dammerer

**Affiliations:** 1Department for Orthopedics and Traumatology, Karl Landsteiner University of Health Sciences, Dr. Karl-Dorrek-Straße 30, 3500 Krems, Austria; 2Division of Orthopaedics and Traumatology, University Hospital Krems, Mitterweg 10, 3500 Krems, Austria

**Keywords:** radiosurgery, spinal metastasis, metastatic spinal disease, spine stereotactic radiosurgery

## Abstract

**Simple Summary:**

Patients with metastatic spine disease face significant challenges and limitations with current therapy options. The aim of this study is to explore the existing literature on spinal stereotactic radiosurgery (SRS) to understand its potential and effectiveness in managing this condition. The motivation stems from a desire to contribute to the field of medical science and improve patient care. This study highlights SRS as a safe and effective technique for managing spinal metastases. It offers good pain control and tumor control with minimal complications. This review strengthens the understanding of SRS for managing spinal metastases, emphasizing its efficacy and the need for personalized treatment plans. Overall, the findings highlight the evolving role of SRS in a multidimensional approach to managing spinal metastases.

**Abstract:**

Background: This study investigated the efficacy of stereotactic radiosurgery (SRS) in managing spinal metastasis. Traditionally, surgery was the primary approach, but SRS has emerged as a promising alternative. Objective: The study aims to evaluate the efficacy of stereotactic radiosurgery in the management of spinal metastasis in terms of local tumor control, patient survival, and quality of life, identifying both advantages and limitations of SRS. Methods: Through an extensive literature search in PubMed with cross-referencing, relevant full-text-available papers published between 2012 and 2022 in English or German were included. The search string used was “metastatic spine diseases AND SRS OR stereotactic radiosurgery”. Results: There is growing evidence of SRS as a precise and effective treatment. SRS delivers high radiation doses while minimizing exposure to critical neural structures, offering benefits like pain relief, limited tumor growth, and a low complication rate, even for tumors resistant to traditional radiation therapies. SRS can be a primary treatment for certain metastatic cases, particularly those without spinal cord compression. Conclusions: SRS appears to be a preferable option for oligometastasis and radioresistant lesions, assuming there are no contraindications. Further research is necessary to refine treatment protocols, determine optimal radiation dose and fractionation schemes, and assess the long-term effects of SRS on neural structures.

## 1. Introduction

Stereotactic radiosurgery (SRS) has gained prominence as a non-invasive treatment for various neurological and oncological conditions. It targets lesions with highly precise radiation, minimizing damage to surrounding healthy tissue [1,2]. This precision is facilitated by magnetic resonance imaging (MRI) and computed tomography (CT) combined with a treatment planning system [3].

SRS has also seen a significant rise in the management of brain tumors, vascular malformations, and pain management. It even shows promising outcomes in treating spine metastasis [2].

Spinal metastasis is the dissemination of cancer to the vertebrae from a primary tumor, which negatively affects a patient’s quality of life, functional abilities, and overall prognosis [4]. Its traditional management (bisphosphonates, chemotherapy, surgery, and conventional external beam radiotherapy (EBRT)) has drawbacks, such as invasiveness and potential radiation-induced toxicity [5,6]. Instead, SRS offers a non-invasive treatment approach, promoting local tumor control while preserving spinal integrity and neurological function, which is achieved in a shorter treatment course as a few or just a single outpatient session [2,7,8,9].

This study aims to elucidate the role of SRS in managing spinal metastases and to inform evidence-based decision-making within clinical practice, undertaking a critical appraisal of the current literature regarding SRS for spinal metastases. This meta-analysis focuses on its efficacy, particularly in regards to pain control, safety aspects, and overall treatment outcomes.

## 2. Materials and Methods

### 2.1. PRISMA Statement

The systematic review followed the recommendations of the Preferred Reporting Items for Systematic Reviews and Meta-Analyses (PRISMA). The protocol has not been registered.

### 2.2. Search Strategy

A systematic literature search was conducted to identify relevant articles published on the use of stereotactic radiosurgery (SRS) for metastatic spine disease with a timeframe of 1 January 2012 to 31 December 2022. PubMed was the primary database and the following search string was used: “metastatic spine disease AND SRS OR stereotactic radiosurgery”. Additionally, an extensive cross-referencing process was employed to maximize the scope of the review to capture potentially relevant studies not identified by the initial search. Articles were initially screened by title and abstract related to diagnostic and treatment principles for SRS in metastatic spine disease.

### 2.3. Selection Process

Studies included in the review were restricted to reviews, recommendations, and guidelines that provide comprehensive discussions on the principles of SRS in managing metastatic spine disease, published in English or German, and providing complete access to their full text to ensure a thorough analysis of the research findings and conclusions.

No statistical methods were used in this literature review.

Table 1 shows the characteristics of all included publications.

Figure 1 represents the study selection process via flowchart.

## 3. Results

A total of 142 studies were initially identified for inclusion. Following a rigorous selection process based on predefined inclusion and exclusion criteria, 21 studies were determined to be eligible for further analysis. The selected studies were analyzed with a focus on pain management, treatment outcomes, and treatment complications.

### 3.1. Pain Management

Stereotactic radiosurgery (SRS) presents a minimally invasive approach for palliative management of spinal metastases, particularly effective for mild spinal cord compression and cancer-related pain control. While both fractionated radiotherapy and SRS are options, the tumor’s close proximity to the spinal cord or cauda equina can compromise the efficacy of fractionated radiotherapy, leading to suboptiomal tumor treatment, which translates to delayed pain relief and increased local recurrence rates compared to SRS [16].

Radiotherapy offers the most benefit 3–4 weeks after treatment, making it most suitable for patients with a life expectancy of at least one month. However, pain relief can be experienced as quickly as after 24 h [8,9]. Accurately predicting life expectancy remains challenging, necessitating a collaborative, multidisciplinary approach [30].

Multiple studies have investigated the application of SRS for pain control in spinal metastasis. These studies consistently demonstrate a high efficacy, with success rates exceeding 90%, a significant achievement even for radioresistant tumors like melanomas in renal carcinoma [25,28].

A study by Gerszten et al. reported that around 85% of patients experience noticeable relief from their symptoms after SRS [31], with 70% finding relief within two weeks and 90% within two months [18].

Supporting these findings, a single-institution study of 500 patients by Gerszten et al. identified pain as the primary reason for treating spinal tumors [16], employing SRS in 67% of the cases (336 patients). Furthermore, SRS demonstrated substantial efficacy in alleviating pain within this challenging patient cohort, leading to an overall enduring pain reduction in 86% (290 patients) of the 336 treated patients, with variations based on the underlying histopathology. The sustained pain improvement was evidenced in 96% of breast cancer cases among women, 96% of melanoma cases, 94% of renal cell carcinoma, and 93% of lung cancer cases [32].

To mitigate the risks of overtreating or undertreating patients, various scoring systems like the revised Tokuhashi, modified Bauer, and Tomita scores are used to estimate life expectancy and guide treatment decisions [33,34,35].

Due to its superior precision and conformal characteristics, SRS is particularly advantageous for patients with oligometastasis or those with specific lesions leading to possible compression of neural elements or cancer-related pain [16,24].

The effectiveness of SRS for both pain management and tumor control has been well established. Notably, patients with isolated vertebral body metastasis without epidural compression are considered ideal candidates for SRS, regardless of their tumor’s histology [30].

However, SRS offers a multifaceted effectiveness in the field of pain management, which can be influenced by the specific tumor location and size, along with the patient’s overall health. Additionally, the potential coexistence of other complications and symptoms can significantly impact SRS’s pain-alleviating outcomes. Therefore, a comprehensive assessment is crucial to determine the potential benefits of SRS in pain management, considering both the characteristics of the tumor and the patient’s overall well-being [16].

### 3.2. Treatment Outcomes

#### 3.2.1. Local Control

This represents the efficacy of a treatment in targeting and managing the tumor at its origin within the spine. This is typically assessed by the success of the SRS in halting or hindering the growth of the metastatic lesion at the irradiated site. Achieving local control means restricting tumor progression within the treated area, reducing its impact on surrounding tissues and potentially alleviating associated symptoms such as pain, compression of neural structures, and instability of the spine.

Early studies investigating initial experience with primary SRS and re-irradiation yielded promising local control outcomes close to 90% effectiveness [36,37,38,39]. However, these findings were based on limited cohorts ranging from 5 to 30 patients. When evaluating larger and more diverse patient cohorts, including about 50% of individuals with a history of prior radiotherapy, the reported success rates of local control have varied between 72% and 90% and 1-year actuarial rates suggests an average local control rate of around 80% [9,40,41].

A retrospective study in a single institution by Gerszten et al. [32] investigated the efficacy of robotic linac-based SRS in a cohort of 500 patients diagnosed with spinal metastases. It consisted of a heterogeneous study population; about 60% presented renal cell (19%), breast (17%), lung (16%), or melanoma (8%) as the primary tumor. Notably, 69% (*n* = 344) had previously received conventional fractionation radiotherapy. The analysis demonstrated an impressive 88% local control rate across the entire group, with some variation depending on the primary histological tumor subtype. Specifically, breast and lung cancers achieved 100% local control, while melanoma and renal cell carcinoma showed lower rates of 75% and 87%, respectively. Although the study provided valuable insights into the outcomes, it did not provide actuarial rates of local control [25].

It is crucial to highlight that achieving local control rates may vary depending on the individual case, underscoring the significance of a multidisciplinary approach involving a team of specialists to decide the most appropriate treatment strategy. Moreover, it is important to acknowledge that these outcomes might not be universally applicable to all patients, as the efficacy of radiosurgery in ensuring local control could be contingent upon diverse variables, such as tumor type, stage, size, location, and the patient’s overall health status [13,19,24,27].

#### 3.2.2. Survival

Spinal SRS primarily offers palliative benefits by improving local tumor control and alleviating symptoms, with patient survival outcomes depending on disease severity, overall health conditions, and primary tumor type [10,15,28,29].

However, studies have shown promising survival rates, alongside notable local tumor control, progression-free survival, and symptom relief. These positive outcomes were observed in patients receiving SRS as an initial treatment, after prior radiation therapy, and following surgical tumor resection [10].

Wowra et al. conducted a prospective interventional case-series study, investigating 102 patients with 134 malignant spinal tumors who underwent single-fraction CyberKnife SRS between August 2005 and October 2007. The study showed a promising local tumor control rate of 98% at 15 months post-treatment and a median survival period post-radiosurgery and primary tumor diagnosis of 1.4 and 18.4 years, respectively [22].

Additionally, the analysis highlighted the Karnofsky performance status (KPS) score [42], a widely used performance scale that assesses the patient’s overall functional status and may influence treatment decisions, primarily in cancer or chronic illness, as an independent predictor of post-radiosurgery survival time [21,22].

Chao et al. [43] conducted a Kaplan–Meier analysis in 174 patients who underwent SRS for spinal metastasis to investigate the prognostic factors for overall survival. They identified three key prognostic factors: time from primary diagnosis (TPD), age, and KPS score. Based on these factors, three prognostic groups were established. The study found significant differences in median overall survival between groups.

Chao et al. concluded that SRS would be most beneficial for patients in group 1 due to their longer life expectancy. Conversely, conventional radiation therapy (CRT) was deemed more appropriate for patients in group 3. Notably, group 2 patients were identified as potential candidates for either SRS or CRT, with SRS being particularly suitable for those with radioresistant histopathological findings, better overall health, or a history of prior therapy [22].

Nevertheless, the studies reviewed highlight that the primary aim of SRS for spinal metastases is to effectively achieve local control and alleviate symptoms like pain and neurological complaints, not to significantly impact overall survival. They suggest that patient mortality is predominantly attributed to the progression of the underlying systemic disease rather than local treatment failure [24].

### 3.3. Dose Regimens

Dose regimens and fractionation strategies for SRS in spinal metastasis exhibit heterogeneity across healthcare institutions. This variability stems from continuous advancements in medical practices, the specific patient’s characteristics, and the evolving technology available at each institution [29]. Consequently, this highlights the importance of an individualized approach to treatment planning, emphasizing meticulous evaluation of patient-specific factors to determine the optimal radiation therapy protocol.

In general, the optimal dose and fractionation regimen for SRS for spinal metastatic disease may vary depending on various individual factors (tumor size, location, proximity to critical structures, number of consecutive vertebral bodies), specific disease characteristics, and physician preference for single- versus multi-fraction regimens [21,23,24,25].

The optimal dose-fractionation regimen for SRS is still under investigation. Studies suggest a range of 10–24 Gy delivered in one to five fractions (six, seven, nineteen), as this approach is believed to effectively relieve symptoms and control the tumor locally. However, recent evidence suggest that higher radiation doses (>24 Gy) might lead to even better local tumor control outcomes [18,22,30].

Short-course regimens could be considered for patients with poor general condition, extensive visceral metastases, and a limited life expectancy (less than 6 months), as they are more cost-effective and less time consuming. Conversely, patients with a life expectancy greater than 6 months and less visceral metastases could benefit from long-course and high-dose regimens [22].

A significant variation in radiation dose (8–30 Gy) was observed in studies focused on epidural metastasis, with distinct lower doses applied for intradural tumors, administered in one to five fractions. Interestingly, Moulding et al. reported poorer local control with lower doses (18–21 Gy) compared to higher doses (18–24 Gy). While some studies employed uniform doses across all patients, others took into account factors like prior CRT, proximity of epidural metastases to the spinal cord, and tumor histology to customize the dose regimen. Furthermore, lower radiation doses were chosen for intradural and intramedullary metastases, as well as for recovery treatment following prior CRT, likely to mitigate the risk of late radiation-induced myelopathy [28,36,44,45,46].

Moulding et al. [47] investigated 21 patients who underwent metastatic spinal cord surgical decompression followed by high-dose single-fraction radiosurgery, with doses varying from 18 to 24 Gy, revealing an overall local tumor control rate of 81%. The study exposed a statistically significant difference in local control rates between patients receiving lower doses (40% control rate in 2 out of 5 patients) compared to those receiving higher doses (93.8% control rate, 15 out of 16 patients). This finding highlights the potential superiority of higher radiation doses in achieving enhanced local tumor control rates for SRS after spinal surgical decompression [22].

Similarly, Yamada et al. [48] investigated 93 patients with spinal metastasis who underwent intensity-modulated radiotherapy (IMRT) between 2003 and 2006 with doses varying from 18 to 24 Gy, revealing a high overall local control rate of 90%, with only 7 patients experiencing treatment failure. Furthermore, the study identified a significant correlation between radiation dose and local control rate, with higher doses (24 Gy or greater) associated with significantly improved outcomes compared to lower doses (*p* = 0.03 for 24 Gy vs. <24 Gy, *p* = 0.04 for >23 Gy vs. <23 Gy) [22].

### 3.4. Treatment Complications

Despite the precision of SRS for a targeted lesion, it can cause collateral damage to surrounding healthy tissue. This radiation-related toxicity is influenced by several factors, including tumor size, location, radiation dose, and treatment regimen, as well as the patient’s overall health status [16,22,24]. This radiation-induced toxicity is typically categorized based on its timing of occurrence as:Acute: refers to toxic reactions emerging shortly after radiation therapy in surrounding tissues and may include symptoms like nausea, vomiting, and radiation-induced esophagitis [21,22,24].Subacute: involves radiation myelopathy, vertebral compression fractures, and bone marrow toxicity. While subacute complications like radiation myelopathy are generally more concerning than reversible acute complications, they are quite rare, have been hardly documented in the literature, and their development time and manifestations often exceed the life expectancy of patients with spinal metastasis [22].Late: manifests long after the radiation treatment and often implies the development of secondary malignant tumors [22].

#### 3.4.1. Spinal Cord

SRS for spinal metastases presents a potential risk of inducing spinal cord and nerve root injury. A recognized complication is delayed radiation-induced myelopathy. While fractionated radiotherapy with 2 Gy per fraction uniformly administered has a low myelopathy risk (<0.5%) at a total dose of 45 Gy, the precise tolerance level for single-fraction SRS of the spinal cord remains unclear. However, a range of 8 to 10 Gy for uniform exposure is estimated [49,50,51].

Gibbs et al. conducted a study identifying a 0.6% incidence (6 out of 1075 patients) of radiation-induced myelopathy. The median time to presentation was 6.3 months after SRS, ranging from 2 to 9 months [52]. While no specific dosage-related factors were identified, it was noted that all patients received doses exceeding 8 Gy. Among all of these patients, one progressed to paraplegia, two remained stable, and three showed symptom improvement.

A study by Ryu et al. [53] provided dosimetric data for partial volume of neural structures based on the treatment of 230 tumors in 177 patients. They used single-fraction radiosurgery, ranging from 8 to 18 Gy, prescribed to the 90% isodose. The spinal cord volume was defined as encompassing an additional 6 mm above and below the radiosurgery target. Their recommendations suggest restricting the spinal cord volume receiving a 10 Gy dose to 10% of the total spinal cord volume, whereas Yamada et al. define Dmax as 14 Gy [20,24].

The impact of SBRT fractionation along different spinal cord segments remains unclear. Conversely, the cauda equina seems to exhibit a higher tolerance to hypofractionated SRS compared to the spinal cord. However, specific SRS doses that may lead to cauda equina injury are not extensively characterized [28].

#### 3.4.2. Vertebral Bodies

SRS and CRT are both reported to cause vertebral compression fractures (VCFs) as a possible adverse effect, especially following de novo SRS treatment [12,14,18]. Myrehaug et al. reported a VCF incidence of 12%, aligning with data from multiple institutions [54]. While many of these fractures remained asymptomatic and did not require surgery, some lead to neurological impairment and intensified pain [18]. To mitigate this risk, multiple studies have explored the incidence and risk factors associated with VCFs post-SRS or -CRT. The presence of substantial lytic disease, prior fractures, administration of high fractional radiation doses (>19 Gy), or a high baseline neoplastic score indicating spinal instability are identified as risk factors. Recognizing patients with any of these factors is essential to proactively implement preventive measures; a comprehensive case discussion involving a multidisciplinary team that incorporates a spinal surgeon is warranted [55].

In a single-center study by Rose et al. [56], the risk of VCF following single-fraction image-guided intensity-modulated SRS for spinal metastases at Memorial Sloan Kettering was explored. Their study found a high prevalence of new or progressive VCFs, affecting 39% (27 vertebrae) of the treated patients. Lytic lesions (with 6–8-fold greater likelihood compared to sclerotic and mixed lesions), tumor location below T-10 (increasing the probability to 4–6 times), and extensive vertebral involvement (>41%) were identified as significant risk factors for fractures. Patients with fracture progression exhibited increased pain scores, and thus, significantly higher narcotic use and functional decline (KPS score). Notably, bisphosphonate therapy, obesity, posterior element involvement, and local kyphosis did not influence fracture risk.

These factors highlight the importance of careful patient selection when considering SBRT, as potential fracture-related instability might favor a surgery-first approach followed by radiation, which could be a more suitable treatment option, in specific cases, compared to SRS alone [24].

### 3.5. SRS Limitations and Contraindications

SRS is not the optimal treatment modality for metastasis diseases involving multiple spinal levels. In these cases, conventional external beam radiotherapy (CER) or even a combination of therapies may be more effective. SRS is more suitable for localized spinal metastases confined to one or two vertebral segments [28,30].

The existing literature has reported several limitations and contraindications associated with SRS. These include spinal instability, large spine or bone compression (>25%), and prior history of SRS with maximal tolerable doses in adjacent spinal cord segments. In these cases, surgical stabilization, decompression, or even chemotherapy might be taken into consideration as potentially more effective treatment options [20,24,57].

Another limitation of SRS highlighted in the literature is the potential development of radiation-induced myelopathy, which is believed to be dose dependent. The risk is higher after re-irradiation, when multiple radiation fractions are administered per day, leading to a cumulative higher dose. However, the overall incidence remains below 1% [24,30,57].

## 4. Discussion

The key findings of this study of stereotactic radiosurgery (SRS) in the management of spinal metastasis encompass pain management, treatment outcomes, and treatment complications, contributing valuable information to the current knowledge in this context.

This study highlights SRS as a highly effective modality for managing pain caused by spinal metastases. Gerszten et al. report exceptional pain control rates exceeding 90%, indicating significant potential for SRS in cancer-related pain management. Furthermore, patients experience prompt and lasting pain relief, often within 2 weeks to 2 months post-treatment, demonstrating the swift therapeutic action of SRS [25,28]. However, the study emphasizes the multifaceted nature of pain management efficacy, recognizing the influence of both the tumor characteristics and the patient’s overall well-being. The incorporation of predictive models like the revised Tokuhashi, modified Bauer, or Tomita scores helps to look for a multidimensional treatment approach, recognizing the complex interaction of medical psychological and environmental factors [33,34,35].

Achieving local control is paramount for good outcomes in spinal metastatic disease. This study meticulously analyzes the complex interplay of factors that influence this. The wide range of reported local control rates (72% to 90%) highlights the pivotal role of tailoring treatment strategies but also the need for nuanced decision making, considering factors such as prior radiotherapy and histological subtypes [9,40,41]. Gerszten et al.’s study provides a real-world lens, revealing an impressive 88% local control rate within a varied patient cohort. This not only highlights the adaptability of SRS, but also underscores its effectiveness, even in challenging cases, showcasing its adaptability in a range of clinical scenarios [25,27].

Spinal metastasis survival outcomes were conventionally viewed through a palliative lens. However, Wowra et al.’s study shows encouraging statistics, with a median survival period of 1.4 years post-radiosurgery, coupled with a local tumor control rate of 98%. Additionally, prognostic factors identified by Chao et al. enable the classification of patients for a more personalized and informed approach to treatment decisions. These findings not only position SRS as a palliative measure, but also hint at its potential to influence survival outcomes in selective cases [22,43].

The field of stereotactic radiosurgery (SRS) is undergoing a continuous evolution in treatment protocols, reflected in the ongoing debate surrounding dose fractionation regimens, which contributes to variations in SRS protocols across healthcare institutions. While proponents advocate for higher radiation doses (>24 Gy) to enhance tumor control, there is a careful balance to be struck between therapeutic benefits and potential complications. This dynamic landscape reinforces the requirement for a bespoke, individualized approach to treatment planning, where factors such as tumor size, location, and the patient’s overall health status are involved in the decision making [18,22,30].

Another important area of focus involves meticulous evaluation of treatment-induced complications, with particular attention given to radiation-related toxicity, particularly concerning spinal cord injuries. In patients with a shorter life expectancy, there should be an emphasis on precision in treatment planning, acknowledgment potential subacute complications. The findings by Gibbs et al. and Ryu et al. provide valuable insights, offering a refined understanding of the safety margins and limitations associated with delivering effective yet safe radiation doses to the spinal cord [52,53].

Vertebral body complications, such as compression fractures, emerge as a potential adverse event post-treatment, particularly in cases involving de novo treatment. Rose et al.’s study identifies high-risk factors, introducing a proactive dimension to risk assessment and careful patient selection. This involves a multidisciplinary collaboration, including spinal surgeons, to delineate the most suitable candidates for SRS. Meticulous selection is crucial, considering factors like the presence of lytic bone disease, prior fractures, and neoplastic scores indicating spinal instability [54,56].

### Review Limitations

A reasonable number of studies [16] were excluded from this systematic review due to the lack of availability of full-text articles.

It might have been difficult to compare studies and results regarding the different study designs, populations, sample sizes, and various treatment protocols in each study.

A further limitation is that we did not compare SRS to other conventional radiotherapy such as carbon or proton therapy.

## 5. Conclusions

This literature review emphasizes spinal stereotactic radiosurgery (SRS) as a promising, safe, and effective technique for managing spinal metastases in a multidisciplinary setting. It represents an innovative approach for precisely delivering potent doses of radiation while safeguarding nearby critical neural structures and organs, in particular the spinal cord. Increasing evidence supports SRS as a primary or adjuvant therapy, particularly for patients without spinal cord compression, demonstrating satisfactory success in pain management and tumor growth limitation while maintaining a low incidence of complications. While conventional radiation therapy (CRT) remains prevalent, SRS presents a reasonable alternative for specific cases, such as oligometastasis, radioresistant metastatic lesions, and/or lesions with prior irradiation, unless contraindicated. Further prospective studies into optimal radiation dose, fractionation, and long-term effects on neural structures are needed to improve evidence quality.

This thorough exploration of treatment outcomes, pain management, dose regimens, and potential complications strengthens the understanding of SRS in managing metastatic spinal disease. The findings affirm its efficacy, highlight the need for a personalized multidisciplinary approach to treatment planning, and underscore the evolving role of SRS in oncological care, showing the multidimensional management of spinal metastases.

## Figures and Tables

**Figure 1 cancers-16-02787-f001:**
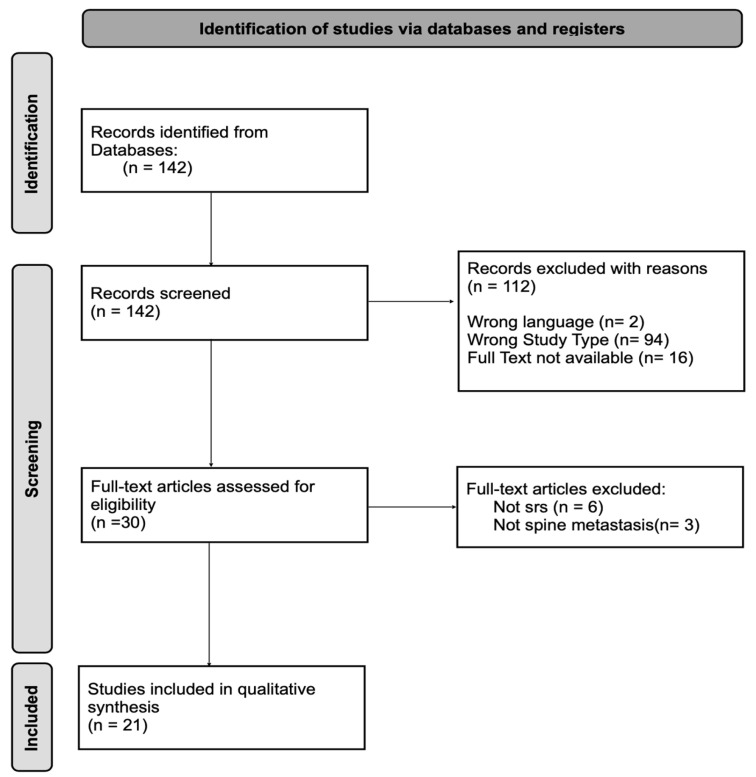
PRISMA flow chart for the literature analysis: it shows the study selection process.

**Table 1 cancers-16-02787-t001:** Detailed study characteristics of all 21 publications included in the review [10,11,12,13,14,15,16,17,18,19,20,21,22,23,24,25,26,27,28,29,30], after a thorough selection.

ID	Study	Year	Region	Country	Study Type	LoE
1	Cox BW et al. [10]	2012	North America	USA	Guideline	n.a
2	Bilsky MH et al. [11]	2014	North America	USA	Review	5
3	Myrehaug S et al. [12]	2017	North America	Canada	Review	5
4	Kurisunkal V et al. [13]	2020	Asia	India	Review	5
5	Caruso JP et al. [14]	2015	North America	USA	Review	5
6	Bowden P et al. [15]	2014	Australia	Australia	Review	5
7	Harel R et al. [16]	2014	Asia	Israel	Review	5
8	Taunk NK et al. [17]	2015	North America	USA	Review	5
9	Bhatt AD et al. [18]	2013	North America	USA	Review	5
10	Fridley J et al. [19]	2019	North America	USA	Review	5
11	Bydon M et al. [20]	2014	North America	USA	Review	5
12	Faruqi S et al. [21]	2022	North America	Canada	Guideline	n.a
13	Zhang HR et al. [22]	2020	Asia	China	Review	5
14	Kotecha R et al. [23]	2020	North America	Canada	Review	5
15	Chawla S et al. [24]	2013	North America	USA	Review	5
16	Jain AK et al. [25]	2014	North America	USA	Review	5
17	Moussazadeh N et at. [26]	2014	North America	USA	Review	5
18	Hadzipasic M et al. [27]	2020	North America	USA	Review	5
19	Joaquim AF et al. [28]	2013	South America	Brazil	Review	5
20	Purvis TE et al. [29]	2017	North America	USA	Review	5
21	Sharan AD et al. [30]	2014	North America	USA	Review	5

LoE: Level of evidence; n.a: not applicable.

## Data Availability

No new data were created or analyzed in this study.

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
