# Peer review of "Stereotactic Radiosurgery in Metastatic Spine Disease—A Systemic Review of the Literature"

_cancers, 2024, doi:10.3390/cancers16162787_

Round 1

Reviewer 1 Report

Comments and Suggestions for Authors

This is an interesting manuscript on a relevant topic.  Subchapter 3.4.2 Vertebral bodies has only the title and the text is missing.

The manuscript lacks a real discussion with other radiotherapic techniques such as conventional radiotherapy, carbon and proton therapy.

Authors should add a paragraph explaining the limitations of SRS (when it is contraindicated and it is better to do something else e.g. surgery)

Author Response

Comments 1: This is an interesting manuscript on a relevant topic.  Subchapter 3.4.2 Vertebral bodies has only the title and the text is missing.

Response 1: Thank you very much for taking your time for the review and for pointing this out. I agree with this comment. Therefore, I have completed the Subchapter 3.4.2 and add the corresponding references.

Comments 2: The manuscript lacks a real discussion with other radiotherapic techniques such as conventional radiotherapy, carbon and proton therapy.

Response 2: We agree. Since the study focuses specially on “Stereotactic Radiosurgery in Metastatic Spine Disease”, as mentioned in the title, we only examined this type of stereotactic radiation therapy and didn’t take in account other conventional radiotherapies like carbon and proton therapy. The aim of the study was to give a systematic literature review on the specific SRS in metastasis spine diseases.

Comments 3: Authors should add a paragraph explaining the limitations of SRS (when it is contraindicated and it is better to do something else e.g. surgery)

Comments 3: Authors should add a paragraph explaining the limitations of SRS (when it is contraindicated and it is better to do something else e.g. surgery)

Response 3: We agree. We added some limitations and contraindications to SRS in the conclusions (subtitle 3.5).

Reviewer 2 Report

Comments and Suggestions for Authors

1)      The title and objective (in the abstract) should be similar. It will help the reader.

2)      The introduction part should be sufficiently informative. It needs to be longer to know the review manuscript's details, keywords, and goals.

3)      The table number should start 1, then 2, and so on. Currently, the first table number is Table 4.

4)      The Figure number should start at 1, then 2, and so on. Currently, the first figure number is Figure 8. One schematic figure can be added.

5)      The Materials and Methods showed that the data were collected from relevant articles published from January 1, 2012, to December 31, 2022. The Results showed that 142 studies were used in a review paper. However, the references indicated that 55 papers were used in this review manuscript. It should be clarified in the revision submission.

6)      The authors could add data/tables supporting search information from PubMed or other search engines.

7)      All figures and table descriptions have to be self-explanatory.

8)      The authors need to check the instructions in this journal. Do they allow bullet points in description writing?

9)      Part 6-Patents could be deleted.

Author Response

Comments 1: The title and objective (in the abstract) should be similar. It will help the reader.

Response 1: Thank you for pointing this out. I have, accordingly, done some changes in the Objectives in the Abstract. “The study sought to evaluate stereotactic radiosurgery for spinal metastasis, aiming to assess local tumor control, patient survival, and quality of life, identifying both its advantages and limitations of SRS”.

Comments 2: The introduction part should be sufficiently informative. It needs to be longer to know the review manuscript's details, keywords, and goals.

Response 2: We agree, that the introduction should be sufficiently informative. In regard of your comment, that the introduction should be longer, we disagree. The introduction mention goals, the aim and the current knowledge given to the literature in short sentences. Furthermore, this study aims to elucidate the role of SRS in managing spinal metastases and to inform evidence-based decision-making within clinical practice, undertaking a critical appraisal of the current literature regarding SRS for spinal metastases.

Comments 3: The table number should start 1, then 2, and so on. Currently, the first table number is Table 4.

Response 3: Thank you for pointing this out. I have changed the table number to 1.

Comments 4: The figure number should start at 1, then 2, and so on. Currently, the first figure number is Figure 8. One schematic figure can be added.

Response 4: I agree. I have changed the figure number to 1.

Comments 5: The Materials and Methods showed that the data were collected from relevant articles published from January 1, 2012, to December 31, 2022. The Results showed that 142 studies were used in a review paper. However, the references indicated that 55 papers were used in this review manuscript. It should be clarified in the revision submission.

Response 5: Thank you very much for the review. The first paragraph of the results explains that a total of 142 studies were indeed initially identified for inclusion. But after a rigorous selection process based on predefined inclusion and exclusion criteria mentioned in the material and methods, only 21 studies were determined to be eligible for further analysis, which are listed in Table 1. The references are the overall literature we used to write this paper.

Comments 6: The authors could add data/tables supporting search information from PubMed or other search engines.

Response 6: We, the authors, are not sure if we understand the comment right. We already included a flowchart to show the search information. If possible, please allow us to ask review 2 to make a more detailed statement or is the flowchart acceptable.

Comments 7: All figures and table descriptions have to be self-explanatory.

Response 7: Thank you for pointing it out. I sum up both descriptions from Table 1 and Figure 1, respectively, for a better explanation.

Comments 8: The authors need to check the instructions in this journal. Do they allow bullet points in description writing?

Response 8: Thank you for review. We agree and have done the proper corrections.

Comments 9: Part 6-Patents could be deleted.

Response 9: Thank you. We have deleted it.

Round 2

Reviewer 1 Report

Comments and Suggestions for Authors

Authors improved the manuscript according to Reviewers requests

Reviewer 2 Report

Comments and Suggestions for Authors

The revised manuscript can be accepted.